# Velociraptor: Leveraging Visual Foundation Models for Label-Free, Risk-Aware Off-Road Navigation

**Samuel Triest**[1], **Matthew Sivaprakasam**[1], **Shubhra Aich**[1], **David D. Fan**[2],
**Wenshan Wang**[1], **Sebastian Scherer**[1]
[1]Carnegie Mellon University Robotics Institute, [2]Field AI
striest@andrew.cmu.edu

**Abstract:** Traversability analysis in unstructured environments is a challenging task that requires understanding of multi-modal inputs such as camera and LiDAR. Measurements from these sensors are often sparse, noisy, and difficult to interpret, particularly in the off-road setting. Existing traversability analysis systems are very engineering-intensive, often requiring hand-tuning of rules and manual annotation of semantic labels. Furthermore, existing methods for analyzing traversability risk and uncertainty are computationally expensive or not well-calibrated. We propose Velociraptor, a traversability analysis system that performs [veloci]ty-informed, [r]isk-[a]ware [p]erception and [t]raversability for [o]ff-[r]oad driving without any human annotations. We achieve this via the use of visual foundation models (VFMs) and geometric mapping to produce a rich visual-geometric representation of the robot's local environment. We then leverage this representation to produce costmaps, speedmaps, and uncertainty maps using state-of-the-art fully self-supervised techniques. Our approach enables intelligent high-speed off-road navigation with zero human annotation, and with about forty minutes of expert data, outperforms several geometric and semantic traversability baselines, both in offline and real-world robot trials across multiple challenging off-road sites.

**Keywords:** Field Robotics, Self-Supervised Learning, Visual Foundation Models

## 1 Introduction

Autonomous off-road driving is a challenging problem in the domain of mobile robotics with applications in many domains such as forestry, defense, exploration and construction [1, 2, 3, 4, 5, 6, 7, 8]. Unlike urban driving and indoor scenarios, off-road terrain is highly unstructured, variable and uncertain. In order to field a high-performing off-road system, it is necessary to consider both geometric and visual information [9]. Unfortunately, the process of transforming this information into representations amenable for planning and control (i.e. traversability analysis) is highly complex.

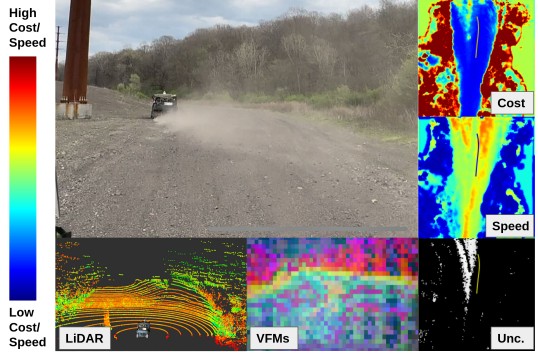

Figure 1: We propose Velociraptor, an off-road traversability system that enables high-speed navigation without requiring any human annotations.

In recent years, there has been much research effort in replacing hand-engineered traversability rules with deep learning [10, 11, 12, 9, 13, 14, 15, 16, 17]. Learning traversability using deep neural networks is promising in that networks can represent complex functions that can directly consume high-dimensional sensory data. However, current approaches generally rely on large hand-annotated datasets [18, 19, 20, 9]. The annotation

8th Conference on Robot Learning (CoRL 2024), Munich, Germany.

process is often laborious, error-prone and brittle to changes in platform or environment, making it challenging and expensive to deploy autonomous mobile robots at scale. Additionally, noisy measurements, novel obstacles and sparse data at high speeds creates the need to capture and quantify uncertainty in order to accurately aggregate information and make risk-informed planning decisions [21, 22, 23, 24]. Especially for self-supervised systems, it is important to be able to both recognize input regions with high epistemic or aleatoric uncertainty and to react accordingly.

We present a method for learning uncertainty-aware traversability from visual and geometric features without semantic annotations or hand-designed traversability rules. This allows us to rapidly deploy our system, requiring only forty minutes of expert driving data and minimal in-field tuning. This is accomplished via the use of geometric analysis and VFMs to produce a rich visual-geometric map representation in bird's eye view (BEV) space for downstream traversability learning. From this, we train a BEV neural network to produce a costmap and speedmap via self-supervised techniques. Given these maps, a model-predictive controller [25] is then used to direct the vehicle to a goal point safely, and at high speed. In our experiments, we find that our system outperforms common geometric and semantic baselines, and that the shared visual-geometric representation is critical to the performance of the self-supervised traversability learning. Relative to our closest point of comparison [11], our work makes the following contributions: **1.** a simple but effective method of combining VFM features and geometric analysis in BEV for traversability learning, **2.** self-supervised learning of costs and *speeds*, allowing for maximum autonomous speeds of around $8m/s$ and **3.** improved uncertainty estimation, allowing for avoidance of out-of-distribution (OOD) obstacles.

## 2   Related Work

The dominant approach for off-road driving leverages semantic segmentation of image and/or Li-DAR data to inform a hand-designed traversability function. Work by Maturana et al. [20] uses point cloud data to project semantic labels from first-person view (FPV) into BEV space. The resulting semantic map is then transformed into a costmap, and is then used by a trajectory library planner for off-road navigation. Additional work by Shaban et al. [12] bypasses the projection step by performing semantic segmentation directly on point cloud data. They leverage this representation for local traversability and planning of a skid-steered robot. This work is extended by Meng et al. [9] to consume image data and output both semantic and elevation information. These methods rely on a hand-designed traversability function to produce costmaps from their semantic representations.

To address the laborious process of hand-annotating semantics, there has been significant interest in self-supervision for traversability. We refer the reader to recent survey papers [7, 26, 27] for a more complete analysis of learning for traversability. Most approaches fall into one of three categories:

**Predicting Future Proprioception:** Methods in this family rely on the fact that there is a correlation between what the robot sees now, and what it will experience in the future. This is exploited by regressing some proprioceptive signal such as slip [13], bumpiness [28, 16, 15] or future trajectory [14, 29] to the perception representation (e.g. images, local maps). This enables the robot to avoid undesirable proprioceptive events (such as high slip or bumpiness) by predicting them from exteroception. Work by Kahn et al. [28] trains a model to predict bumpiness from first-person images and a sequence of actions. This model is trained from random exploration data and enables preferences for smoother terrain such as sidewalks. Work by Wellhausen et al. [24] leverages a ground reaction force to enable legged robots to avoid challenging terrain from vision. Further work [16, 15] performs this prediction directly in BEV and demonstrate effectiveness on wheeled and legged platforms.

**Imitation of Experts:** Methods in this family typically rely on inverse reinforcement learning (IRL) [30, 31] to learn a cost function under which a set of expert data is made optimal. Pioneering work by Ratliff et al. and Bagnell et al. [32, 33] presents an application of IRL for global navigation in which costmaps are learned from satellite imagery and applied to a large-scale mobile robot. Wulfmeier et al. [34, 35] demonstrate that the gradient of the MaxEnt IRL objective can be used to train deep

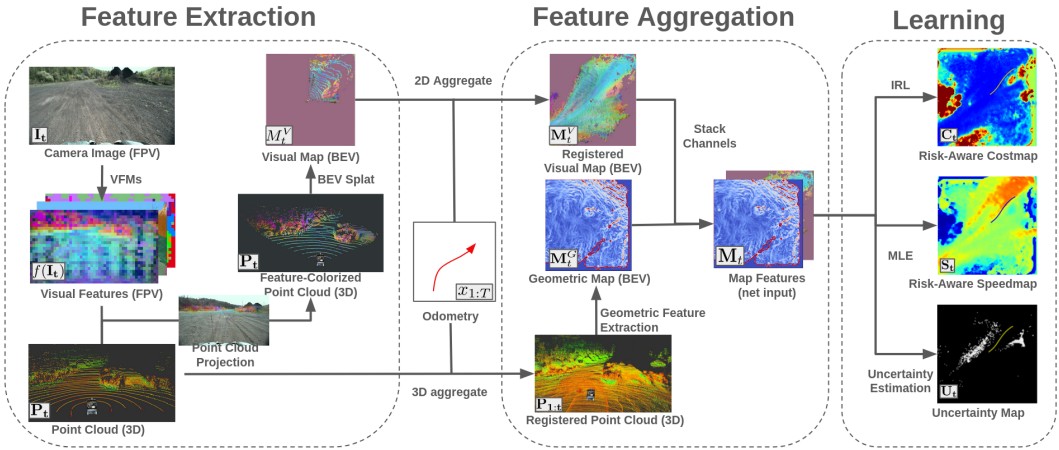

Figure 2: A high-level overview of our perception pipeline. At every timestep, the robot receives a camera image, point cloud and state estimate. The image is first run through a deep visual feature extractor to produce image-space features. These features are used to decorate the point cloud and are BEV-splatted to produce a single-frame visual map. These maps are aggregated over time using odometry. Concurrently, the pointclouds themselves are registered using odometry and have BEV geometric features extracted. Finally, the geometric and visual maps are stacked together and fed through a FCN to generate our costmaps/speedmaps/uncertainty.

neural networks, and demonstrate its efficacy in semi-structured urban driving. Additional work applies MaxEnt IRL to local navigation for wheeled and legged robots [36, 37, 38, 11].

**Learning Model Uncertainty:** Additional work leverages model uncertainty as a proxy for traversability. Work by Fan et al. [39] (STEP) presents a LiDAR-based traversability pipeline that reasons about *distributions* of cost. Uncertainty is propagated through hand-engineered traversability rules, and is transformed into scalar cost using Conditional Value-at-Risk (CVaR) [40]. Work by Schmid et al. [41] leverages the uncertainty of a learned perception model as a metric of traversability. Additional work by Cai et al. [42] learns a perception-conditioned traction distribution. This is leveraged by a model-predictive controller to enable risk-aware planning via dynamics rollouts that sample from this learned distribution. This work is extended [21] to include estimation of epistemic uncertainty and demonstrations on both wheeled and legged platforms.

**Leveraging VFMs:** There has been recent interest in leveraging VFMs for off-road traversability. Existing work V-Strong [29] leverages SAM [43] for contrastive learning of traversability in image space. Existing work WVN [13] leverages Dino [44] to regress a proprioceptive slip estimate onto FPV embeddings. These estimates are then projected onto an elevation map for BEV-space traversability and demonstrated on a legged robot. We differ from prior work in that we first map VFM features into BEV, and then perform learning. This simplifies the process of learning multiple outputs, enables simple fusion with geometric LiDAR information and reduces the effect of the lossy FPV to BEV projection step. Furthermore, we draw from all three families of self-supervised traversability by **1.** learning to predict future expert speeds from our exteroceptive map, **2.** learning to predict cost from expert demonstrations via IRL, and **3.** performing epistemic and aleatoric uncertainty estimation on our perception representation.

## 3 Visual-Geometric Mapping with VFMs

It is necessary to produce a perception representation for downstream control that enables a robot to navigate to desired goal locations safely and quickly, without the use of any prior information (e.g. road maps, satellite imagery). For this work, we will use a local grid map as this representation [45], which we will populate with information derived from image and point cloud observations. At a

high level, our method produces local maps for learning, given streams of state estimates $x_t$, point clouds $\mathbf{P}_t$ and images $\mathbf{I}_t$. First, we maintain an aggregated point cloud $\mathbf{P}_{1:t}$ via registration from our state estimates $x_{1:t}$. Then we extract BEV-space geometric features (e.g. elevation, slope, etc.) from this registered point cloud to produce a geometric map $\mathbf{M}_t^G$. In parallel, we compute deep visual features from the current image $\mathbf{I}_t$ using VFMs. The current point cloud $\mathbf{P}_t$ is projected into the image frame to assign spatial positions to the deep image features. These features are then projected into BEV space and aggregated over time to produce a visual map $\mathbf{M}_t^V$. Finally, these two maps are stacked to yield a visual-geometric map $\mathbf{M}_t$. This process is shown in Figure 2. A summary of each component of the mapping system are presented below, with additional details in the supplemental.

## 3.1   Obtaining Geometric Features

In order to obtain geometric features, we register the pointclouds $\mathbf{P}_{1:t}$ together using odometry $x_{1:t}$. Then given a local map size and resolution, we perform a series of geometric analyses to extract several salient features (such as terrain height, slope, etc.). Note that this geometric analysis leverages a registered point cloud. This is preferable to computing single-frame maps and aggregating, as the computation of several geometric features is influenced by effects such as occlusion, which aggregated point clouds can mitigate somewhat.

## 3.2   Obtaining Visual Features

Visual foundation models such as Dinov2 [46] and SAM [43] represent an exciting opportunity for the off-road driving domain in that they provide continuous-valued, semantic-level features for a wide range of image data without requiring fine-tuning or annotation. However, it remains an open question as to how best to incorporate such information into a navigation stack. We provide a baseline approach for incorporating visual features based on LiDAR point cloud projection that has been used for semantics in prior work [20]. Additional details are provided in the supplemental.

In order to improve memory efficiency and speed of mapping, we perform a principal component analysis (PCA) on the FPV VFM images in the train set and map the top $n$ components. This is desirable because it maintains the most relevant features from the VFM, while significantly reducing the number of feature channels and being efficient to compute. We then project the current LiDAR point cloud $\mathbf{P_t}$ into the image frame to assign spatial positions to the reduced VFM features. Features with a lidar return are then mapped into BEV to produce a visual map (cells containing multiple points use the average embedding), which is aggregated over time with odometry.

An important point to note is that unlike the geometric mapping, the aggregation of visual features occurs in map-space (as opposed to projecting the aggregated point cloud into the current image). This is done to avoid the effect of occlusions and odometry error, which can occur when points are projected into images at different timesteps. Additionally, this process qualitatively reduces the effects of positional embedding artifacts on the aggregated map.

## 4   Learning in BEV from Unlabeled Data

We can leverage our BEV representation to perform several label-free traversability tasks. In order to provide the most useful guidance to a downstream controller, we elect to produce three maps, a speedmap $\mathbf{S}_\theta$, costmap $\mathbf{C}_\theta$ and uncertainty map $\mathbf{U}$ (Figure 2). The costmap informs the vehicle of *where* to drive by producing a single scalar value commensurate to the difficulty of traversal over that particular cell (i.e. trails have low cost, trees have high cost). The speedmap informs the vehicle of a notion of *how* to drive in that for every cell by producing a recommendation of how fast a cell can be safely traversed over (i.e. trails have high speeds, and tall vegetation has low speeds). The uncertainty map identifies unseen terrain features which should be avoided. In order to learn these maps, we use a lightweight fully-convolutional network (FCN) with a shared backbone and separate

heads for costmap and speedmap prediction. Following prior work [32, 11], we train an ensemble of these networks in order to perform uncertainty estimation.

## 4.1 Learning Costmaps

Unlike geometric and semantic features, there is no obvious heuristic mapping from VFM features to cost. Thus, in order to learn costmaps, we leverage inverse reinforcement learning (IRL). IRL is desirable in this application due to its demonstrated success in off-road traversability [33, 32, 35, 37, 11] and ability to consume arbitrary state features, which need not be human-interpretable.

At a high level, IRL seeks to learn a cost function under which a corpus of expert demonstrations is made optimal. We optimize an objective similar to LEARCH [32], in which positive samples are generated from the expert trajectory, and negative samples are generated from a fast trajectory optimizer (in our case, MPPI [47]). State visitation distributions $D^E$ and $D^L$ (discretized over grid states $\mathcal{S}$ in the map) are computed for the expert and learner, and used via Equation 1 to update the network. We follow the example set by Ratliff et al. [32] and train our network to output log-costs, which we exponentiate for the planner. We also use a small L2 regularizer on costs.

$$\nabla_\theta \sum_{\tau_E \in \mathcal{D}_E} \mathcal{L}(\tau_E|\theta) = \sum_{s_i \in \mathcal{S}} \left[ (D^E_{s_i} - D^L_{s_i}) \frac{\partial \mathbf{C}}{\partial \theta} \right] + k \sum_{s_i \in \mathcal{S}} \mathbf{C}_\theta(\mathbf{M})_{s_i}, \quad \mathbf{C}_\theta(\mathbf{M}) = exp(f_\theta(\mathbf{M})) \quad (1)$$

## 4.2 Learning Speedmaps

In order to learn speedmaps, we perform supervised learning of the expert speeds, conditioned on our perception representation. We follow the example of Cai et al. [21] by learning a categorical distribution over discrete speed bins, per cell. This has the advantage of correctly representing the support of the speed variable ($\mathbb{R}^+$), while maintaining the ability to represent arbitrary distribution shapes. Learning a distribution allows us leverage quantiles of this learned speed distribution as a risk tolerance parameter. This distribution is learned by maximizing the log-probability of the expert speed $v(x)$ for its corresponding map cell $s(x)$. For safety purposes, we also introduce negative samples for cells where the expert did not drive and set their speed label to zero, resulting in the objective in Equation 2 (where $\mathbf{S}^s_\theta$ denotes the current learned speed distribution at map state $s$).

$$\mathcal{L}_{speed} = \sum_{\tau_E \in \mathcal{D}_E} \left[ \sum_{x_i \in \tau_E} -log\mathbf{S}^{s(x_i)}_\theta(||v(x_i)||) + k \sum_{s_i \in \mathcal{S} \setminus \tau_e} -log\mathbf{S}^{s_i}_\theta(0) \right] \quad (2)$$

## 4.3 Learning Uncertainty Maps

We also perform density estimation of our training input distribution to detect regions of high epistemic uncertainty. We observe that the IRL network is effective at detecting distinct geometric obstacles and evaluating the traversability of different types of terrain, and our offline results indicate generalizability to held-out test sites. However, we noticed that various out-of-distribution objects that don't resemble terrain didn't trigger high-cost predictions from the network as expected. This is undesirable, as in practice we found some of these objects to be dangerous (e.g. a stray tire or a pallet full of nails). While this may not be the case every time, within the scope of this work we adopt a policy of avoiding these objects if the robot can find low-cost terrain around it.

Offline, we randomly sample image embeddings from the train dataset, and use K-Means clustering to generate feature clusters (using cosine distance). Online, we leverage the residuals between the features and cluster centers and perform the following operation to determine uncertainty $U$.

$$u = \min_{k \in \{1,2,\dots,K\}} ||d - F_k||_1, \quad U = \begin{cases} 0 & \text{if } u < \tau_U \\ u & \text{otherwise} \end{cases} \quad (3)$$

This operation is done in the map space after the visual features have already been projected out. If the minimum distance of a map feature $d$ to any cluster $F_k$ exceeds a threshold $\tau_U$, we assume its corresponding terrain/object is uncertain and therefore assign it a high cost, allowing us to avoid anomalies without any further training.

## 4.4  Risk-Awareness

Since we are able to produce distributions of both cost and speed, we are able to adapt the risk-tolerance of our learned maps. Note that both our speed and cost distributions can be non-Gaussian and multi-modal. In order to produce a risk-aware cost, we follow prior work [39, 11, 48] and use Conditional Value-at-Risk (CVaR) [40]. In order to produce a risk-aware speed limit, we simply take the $\nu$-th quantile of the learned speed distribution per cell (Equation 4).

$$C_\nu(\mathbf{M}) = \text{CVaR}_\nu(\{\mathbf{C}_{\theta_i}(\mathbf{M}), \forall i \in E\}), \quad S_\nu(\mathbf{M}) = \text{VaR}_\nu(\{\mathbf{S}_{\theta_i}(\mathbf{M}), \forall i \in E\}) \tag{4}$$

## 5  Experiments and Analysis

### 5.1  Offline Tests

We first evaluate our method on several held-out datasets. The first dataset is a representative sample of expert driving from the test site where the training data was collected (Dataset 1). The second dataset (Dataset 2) was collected from a different test site (approx. 70 miles away) and contains higher speeds, more slopes and denser vegetation than Dataset 1.

In order to evaluate our learned control representation, we compare the optimal trajectory under MPPI to the expert trajectory (Equation 5, cost function $J$ in supplemental). We use Modified Hausdorff distance on positions (pMHD), speeds (sMHD), and both positions and speeds (MHD). This metric is standard in prior work for costmap prediction [36, 37, 11, 9]. We report positional and speed distances separately to somewhat (though not completely) separate out the contribution of the costmap and speedmap. We compare our method to several strong baselines in recent literature.

$$\mathcal{L} = MHD(\tau_E, \tau^*), \quad \tau^* = \min_\tau J(\tau, C_\nu, S_\nu) \tag{5}$$

**Baseline 1 - Geometric Analysis from ALTER [49]:** We chose the cost function from ALTER as a geometry-only baseline. At a high level, ALTER weights a baseline occupancy map with SVD decompositons of the points in each cell to encourage the vehicle to drive on smooth terrain when possible. Since ALTER does not produce a speedmap, linear quantile regression was performed on the train set to predict expert speeds from this SVD feature. We use two variants of the ALTER cost function, one which uses the coefficients directly from Chen et al. [49] (ALTER orig), and one with modified coefficients that yielded better on-platform navigation performance (ALTER tuned).

**Baseline 2 - Semantics from GANav [50]:** We chose GANav as a representative semantic baseline. The FPV semantic predictions from GANav were then passed through our visual mapping pipeline in order to generate BEV semantic maps. Note that we follow the approach of Asgharivasi et al. [51] and map the semantic logits (which we then softmax in the aggregated map). These maps were then transformed into costs via a hand-designed reward function (which was validated and tuned to give good navigation performance on platform). Speeds were computed by computing the class-conditioned speed quantiles on the train set.

**Baseline 3: Simple Visual-Geometric Fusion (ALTER + GANav):** We also evaluate a simple fusion of geometric and visual costs. This is achieved by combining the previous baselines via a cell-wise maximum for costs, and minimum for speeds.

In addition to the above baselines, we evaluate four variants of Velociraptor. This is done to ablate the usefulness of various visual features for the downstream navigation task. **V-geom** is trained

Table 1: Results for cost CVaR = 0, speed quantile = 0.5

|  | pMHD (1) | sMHD (1) | MHD (1) | pMHD (2) | sMHD (2) | MHD (2) |
|---|---|---|---|---|---|---|
| ALTER cfn orig [49] | $1.62 \pm -$ | $0.31 \pm -$ | $1.98 \pm -$ | $2.99 \pm -$ | $0.51 \pm -$ | $3.37 \pm -$ |
| ALTER cfn tuned [49] | $1.33 \pm -$ | $0.25 \pm -$ | $1.65 \pm -$ | $3.36 \pm -$ | $0.52 \pm -$ | $3.70 \pm -$ |
| GA-Nav cfn [50] | $3.70 \pm -$ | $0.43 \pm -$ | $4.01 \pm -$ | $4.24 \pm -$ | $0.76 \pm -$ | $4.70 \pm -$ |
| ALTER orig + GA-Nav | $3.14 \pm -$ | $0.42 \pm -$ | $3.47 \pm -$ | $4.63 \pm -$ | $0.67 \pm -$ | $4.97 \pm -$ |
| ALTER tuned + GA-Nav | $2.56 \pm -$ | $0.37 \pm -$ | $2.88 \pm -$ | $3.89 \pm -$ | $0.76 \pm -$ | $4.37 \pm -$ |
| V-geom [11] | $1.26 \pm 0.18$ | $0.23 \pm 0.04$ | $1.46 \pm 0.19$ | $2.34 \pm 0.27$ | $0.37 \pm 0.03$ | $2.65 \pm 0.28$ |
| V-semantics | $1.16 \pm 0.08$ | $0.20 \pm 0.01$ | $1.35 \pm 0.08$ | $2.26 \pm 0.19$ | $0.46 \pm 0.04$ | $2.63 \pm 0.21$ |
| v-Dino | $\mathbf{0.97 \pm 0.03}$ | $\mathbf{0.18 \pm 0.00}$ | $\mathbf{1.15 \pm 0.03}$ | $\mathbf{1.82 \pm 0.08}$ | $\mathbf{0.30 \pm 0.02}$ | $\mathbf{2.10 \pm 0.09}$ |
| V-SAM | $1.13 \pm 0.10$ | $0.20 \pm 0.01$ | $1.41 \pm 0.10$ | $2.12 \pm 0.24$ | $0.34 \pm 0.03$ | $2.42 \pm 0.25$ |

Table 2: Results for cost CVaR = 0, speed quantile = 0.9

|  | pMHD (1) | sMHD (1) | MHD (1) | pMHD (2) | sMHD (2) | MHD (2) |
|---|---|---|---|---|---|---|
| ALTER cfn orig [49] | $1.15 \pm -$ | $0.20 \pm -$ | $1.44 \pm -$ | $1.57 \pm -$ | $0.17 \pm -$ | $1.78 \pm -$ |
| ALTER cfn tuned [49] | $1.06 \pm -$ | $0.17 \pm -$ | $1.32 \pm -$ | $1.99 \pm -$ | $0.16 \pm -$ | $2.16 \pm -$ |
| GA-Nav cfn [50] | $3.37 \pm -$ | $0.46 \pm -$ | $3.66 \pm -$ | $2.97 \pm -$ | $0.30 \pm -$ | $3.21 \pm -$ |
| ALTER orig + GA-Nav | $2.68 \pm -$ | $0.39 \pm -$ | $2.97 \pm -$ | $2.96 \pm -$ | $0.27 \pm -$ | $3.15 \pm -$ |
| ALTER tuned + GA-Nav | $2.27 \pm -$ | $0.31 \pm -$ | $2.55 \pm -$ | $2.02 \pm -$ | $0.20 \pm -$ | $2.23 \pm -$ |
| V-geom [11] | $1.05 \pm 0.13$ | $0.18 \pm 0.01$ | $1.31 \pm 0.13$ | $1.64 \pm 0.12$ | $0.16 \pm 0.01$ | $1.82 \pm 0.12$ |
| V-semantics | $1.11 \pm 0.11$ | $0.17 \pm 0.02$ | $1.34 \pm 0.08$ | $1.55 \pm 0.07$ | $0.17 \pm 0.01$ | $1.74 \pm 0.07$ |
| V-Dino | $\mathbf{0.83 \pm 0.03}$ | $\mathbf{0.14 \pm 0.00}$ | $\mathbf{1.08 \pm 0.02}$ | $\mathbf{1.48 \pm 0.02}$ | $\mathbf{0.15 \pm 0.01}$ | $\mathbf{1.66 \pm 0.03}$ |
| V-SAM | $1.01 \pm 0.13$ | $0.16 \pm 0.02$ | $1.25 \pm 0.13$ | $1.59 \pm 0.13$ | $\mathbf{0.15 \pm 0.01}$ | $1.77 \pm 0.13$ |

with only geometric features, which is equivalent to Triest et al. [11] with the addition of the extra head for speed. **V-semantics**, **V-Dino** and **V-SAM** are trained with both geometric features and the features from GANav [50], DINOv2 [46] and SAM [43], respectively. All other training parameters for these models are held constant.

Quantitative results are provided in Tables 1 and 2. In Table 1, the speed risk level (quantile of speed distribution) $\nu_s$ is set at the mean value (0.5) for all methods. For Table 2, $\nu_s$ is set to 0.9. We report the mean and standard deviation over five trials for learning-based methods.

We find that it is important to learn the cost and speedmaps as opposed to relying on existing heuristics, as evidenced by significant improvement based on the metrics. As expected, the addition of visual features allows for improved trajectory matching, with the features from Dinov2 appearing to be the most useful, based on superior performance across all metrics and datasets. As one might expect, considering distributions of cost and speed are important, especially given that speed is treated as an upper bound. This is evidenced by improvement across all methods when $\nu_s = 0.9$.

## 5.2 Large-Scale, High-Speed Tests

We evaluated our method and several baselines on a $4km$ loop in an off-road testing site. This loop contains many challenging traversability scenarios such as steep slopes, tall grass, standing water, and small, OOD obstacles. We measure performance via the number of safety operator corrections, average autonomous speed and speed-normalized ride bumpiness [16] (Corrections (long), Avg.

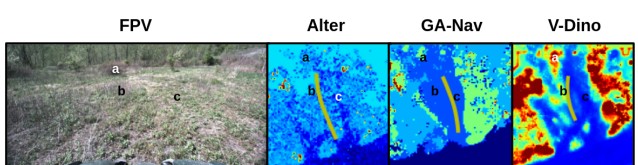

Figure 3: Leveraging fine-grained visual features allows for more sophisticated costing of different types of grass and vegetation. We can observe that our method can distinguish between low shrubs (a), marshy grass (b), and low, traversable grass (c) where geometric and semantic baselines cannot. Planned trajectories are shown in yellow. Costmaps are cropped to focus on the relevant region.

Speed and Avg. Bump in Table 3). For safety, experiments were run with more conservative risk parameters than the offline test ($\nu_c = 0, \nu_s = 0.25$ for learned, and $\nu_s = 0.5$ for baselines).

Overall, we find that our learned costmap and speedmap enabled the robot to complete the course with fewer operator interventions at higher speeds. In particular, visual features were important for allowing the vehicle to travel more quickly on open trails. A combination of geometry and continuous-valued visual features allowed for disambiguation of cost within a semantic class (Figure 3). We found that all methods achieved a similar amount of ride bumpiness. This is likely due to

Table 3: Hardware Results

| Method | Corrections (long) | Avg. Speed | Avg. Bump | Corrections (short) |
|---|---|---|---|---|
| Alter cfn tuned [49] | 10 | 4.64m/s | 0.47 | 5 |
| GA-Nav [50] | 15 | 3.93m/s | 0.48 | 5 |
| V-Dino | **6** | **5.06m/s** | 0.47 | 5 |
| V-Dino + Unc | 10 | 4.59m/s | 0.48 | **1** |

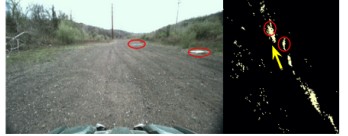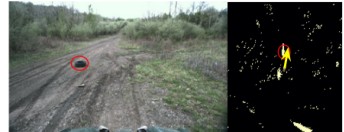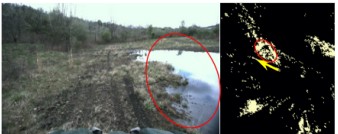

Figure 4: Example scenarios with geometrically-indistinct objects (tarps, tire, puddle) obstructing the path in red (robot pose is yellow arrow). The uncertainty layer classifies them as OOD objects.

the resolution of the costmap, and the lack of explicit bumpiness minimization in our cost function. Several additional scenarios are presented in the supplemental.

## 5.3 Avoidance of OOD Objects

To isolate the effect of the OOD detection, we designed a smaller course where the robot had to navigate to a goal directly straight ahead, with various OOD obstacles (two tires, a flattened sign, and a tarp) placed in its path. We evaluate the number of corrections for each method over three trials, the results of which are shown in the Corrections (short) column of Table 3 (note that the short and long trials are separate). We find that the uncertainty layer adds additional cost to these obstacles without adding cost to previously seen terrain like trails and tall grass (Figures 4, 5).

## 6 Limitations and Future Work

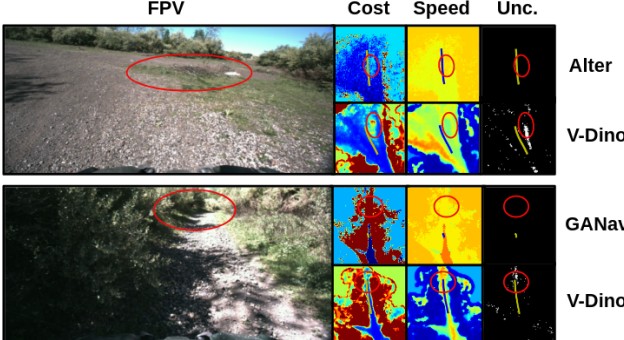

Figure 5: Examples from our large-scale navigation trial. Relevant obstacles are circled in red for FPV and BEV. In the top scenario, the robot must navigate around a short pile of debris that incurs a risk of tire puncture. The geometric method fails because of the low height of this obstacle. In the bottom scenario, the robot must navigate through a trail that necessitates contact with with dense, overhanging vegetation. The semantics-based method is overly conservative and does not plan through. In both cases, our method succeeds by learning from geometry and vision together.

While our method represents a promising step towards fully self-supervised, adaptive traversability, there are a number of limitations that must be addressed in future work. First, our method does not address occlusions and consistency of features at different distances. This affects both the visual and geometric features, which can affect downstream traversability. We expect that applying existing inpainting methods [52, 12, 9, 17] as well as longer-range planning, can alleviate this problem while remaining self-supervised. Second, our method of aggregating and projecting VFM features to BEV, while effective, is overly simple. We plan to extend our mapping procedure to more expressive map representations such as voxels. Lastly, our representation output is limited in its ability to provide fine-grained trajectory guidance. While we are able to provide a maximum speed and cost, we cannot produce certain behaviors such as approaching logs from certain angles, or going at a minimum speed. We can ameliorate this by incorporating physics-informed costs in our MPPI cost function [16, 53, 54], or performing learning on trajectory-level features such as speed and curvature.

**Acknowledgments**

The authors would like to thank Amirreza Shaban, Calvin Chung, Ali Agha and Cherie Ho for their advice and insights during the paper writing and review phases, and Ian Higgins for help in capturing drone footage of our hardware experiments. This work was partially supported by Field AI, Inc. Research was sponsored by the Army Research Laboratory and was accomplished under Cooperative Agreement number W911NF-21-2-0152 and W911NF-18-2-0218. The views and conclusions contained in this document are those of the authors and should not be interpreted as representing the official policies, either expressed or implied, of the Army Research Laboratory or the U.S. Government. The U.S. Government is authorized to reproduce and distribute reprints for Government purposes notwithstanding any copyright notation herein.

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
