# OpenReview forum: "Velociraptor: Leveraging Visual Foundation Models for Label-Free, Risk-Aware Off-Road Navigation"
_robot-learning.org/CoRL/2024/Conference — CoRL 2024_

### Official Review · Reviewer_yWjL · 2024-07-19

**Originality:** 4
**Technical Quality:** 4
**Clarity Of Presentation:** 5
**Potential Impact:** 3
**Recommendation:** 4
**Confidence:** 3

**Review:**

**Summary:**

Overall, I think the proposed method effectively combines Visual Foundation Models (VFMs) with geometric features and generates high-quality maps for off-road navigation. The paper is clearly-written, and the experiment results are convincing. So I would recommend this paper to be accepted.

**Disclaimer:**

While I have some knowledge in off-road navigation, I'm not very familiar with related work and/or state-of-the-art methods. So my evaluation of this paper might not be highly accurate.

**Strengths:**

* Effective combination of geometry and semantics using VFMs

This paper presents an effective method that combines multiple sensory inputs (RGB camera, Lidar) to learn self-supervised cost and speed maps for offroad navigation. The use of VFM leads to higher-quality maps with finer granularity feature extraction (Fig.4-5), and leads to improved real-robot performance (Table.3).

* Well-designed experiment that demonstrates the effect of VLM

Overall, I found the paper's experiments to be well-designed with well-designed baselines and insightful analysis. I particularly like Fig.4 and 5, which provides valuable insights on leveraging foundation models for robotics.

* Clearly written and well-organized

I find the paper to be well-organized and enjoyable to read. The main method is clearly outlined in Fig.2 and adequately described in corresponding sections, with further details available in the appendix. This is one of the few papers that allow me to understand the entire methods in "one pass" without back-and-forth.

**Weaknesses:**

I cannot identify any significant weakness of the paper. However, as stated in the disclaimer, I'm not very familiar with related works and SotA methods in this area. Therefore, I might have overlooked important aspects of the paper.

**Quality Of The Limitations Section:**

3

**Questions For Rebuttal:**

I have a few high-level questions for the author:

1. The proposed method uses self-supervised learning where the only "label" comes from human demonstration. How sensitive is it to the quality of human demonstration? Would it be beneficial to further incorporate other information (e.g. body oscillations, slipping) into this framework?

2. How would you extend this framework to a different mobile robot platform (e.g. another off-road vehicle of different dimensions, a legged robot, a wheel-legged robot)?

**Robotics Focus:**

4

**Summary Of Paper:**

The paper presents a framework that combines visual foundation models with geometric mapping to learn maps for high-speed off-road navigation in a self-supervised manner without human annotations.

**Summary Of Recommendation:**

Overall, I think this paper is high-quality and clearly written, and should be accepted.

---

### Official Review · Reviewer_K4L2 · 2024-07-19
**Good paper but lacks novelty**

**Originality:** 2
**Technical Quality:** 3
**Clarity Of Presentation:** 4
**Potential Impact:** 2
**Recommendation:** 2
**Confidence:** 4

**Review:**

Strengths

1. The paper is in general well-written and easy to follow.
2. The maps are learned without manual labeling, hence reducing the data annotation and tuning efforts.
3. Both offline and real-robot experiments were conducted. The results show clear advantages of the proposed system over the baselines.

Weaknesses

1. IRL, speedmap and uncertainty estimation already exist in other off-road driving works, so the novelty aspect is limited.
2. The way how features are projected (simple averaging) is not optimal since a map cell may contain multiple semantic objects or overhanging objects.
3. Some equations (Eq 1, 2, 3) contain symbols that are never introduced, making it difficult to understand them.
4. The dataset is rather small (<3000 samples), which may cause the model to overfit.
5. The uncertainty estimation is mostly a heuristics. It will be better if the paper takes a more principled approach for uncertainty estimation.

Comments

1. While GA-Nav serves as the semantics-only baseline, it is unclear if its inferior performance is due to the lack of additional modality or a less capable vision backbone. Same applies to ALTER.
2. The cost function (Eq. 7 in the supplementary material) is not a weighted sum of the three maps, which contradicts the text.

**Quality Of The Limitations Section:**

3

**Questions For Rebuttal:**

Questions for rebuttal

1. In Table 3, why does V-Dino+Unc performs worse on the long run?
2. What is the motivation of using a heuristics-based uncertainty estimation method? How good is it in terms of false positives? I understand it is difficult to have a quantitative analysis, but it will be informative if you can show a video of a long run visualizing the three maps.
3. What are the causes of the interventions?

**Robotics Focus:**

4

**Summary Of Paper:**

This paper proposes a system for self-supervised traversability estimation in off-road environments. The system uses past LiDAR scans and images to build a aggregated terrain feature map around the vehicle. In particular, geometric features are computed from aggregated LiDAR point cloud, and image features are extracted from Vision Foundation Models. The features are projected into BEV space by averaging, which are then decoded into costmap, speedmap and uncertainty maps. Experimental results show that the system outperforms geometric-only, semantic-only, and simple fusion baselines.

**Summary Of Recommendation:**

The paper is in general convincing but I would like to see more novelty in system design and implementation.

---

### Official Review · Reviewer_Wz3t · 2024-07-21
**Velociraptor: Leveraging VFMs for Label-Free, Risk-Aware Off-Road Navigation**

**Originality:** 3
**Technical Quality:** 3
**Clarity Of Presentation:** 3
**Potential Impact:** 2
**Recommendation:** 3
**Confidence:** 4

**Review:**

The manuscript provides a well-defined and valid methodology for the prediction of cost map based on inverse reinforcement learning and prediction of additional speed and uncertainty maps to enable autonomous off-road navigation at high speeds, using self-supervised learning.
Strengths:
+ The manuscript is clearly written and well organized. The problems are formulated clearly.
+ Evaluation and experiments are demonstrated on a real vehicle/robot.
+ Ablation study covers the important aspects of the usage of VFMs method.

Weaknesses:
- The manuscript lacks analysis of the training and test data distribution and generalizability of the method on unseen terrains.
- Related works didn’t well cover off-road traversability learning research.
- Only a conceptual pipeline is presented as Fig. 2. It would be needed to have a method diagram to show the method merit, rather than only showing the usages and comparison of various VFMs for this application.
- 4.3 Learning Uncertainty Maps lack details of how uncertainty is represented, and does it propagate between frames etc. This subsection is underdeveloped.

Minor weaknesses:
- A few typos and syntax issues, such as: what is OOD?

**Quality Of The Limitations Section:**

2

**Questions For Rebuttal:**

As mentioned in the weaknesses, mainly, the presentation of the method details can be enhanced and method generalization to various terrain discussion would be preferred. More relevant related-work survey. Additionally, without code release, quantitive analysis and model runtime evaluation for downstream real-world robot tasks are hard for research community benchmark.

**Robotics Focus:**

4

**Summary Of Paper:**

The manuscript introduces VELOCIRAPTOR, a method to learn traversability estimation in a self-supervised manner for autonomous navigation of off-road vehicles, which combines visual-foundation models and geometric data in vehicle's Bird's Eye View (BEV). The method also predicts uncertainty estimation along with cost map and speed map which is further utilized by the low level planner and controller. The proposed method doesn't require any human-annotated labels and hand engineering of the traversability estimation function, which makes this method suitable for the application of autonomous off-road navigation.

**Summary Of Recommendation:**

The manuscript is well organized and comprehensive. It provides cool evaluation demo for the application of off-road autonomous navigation. It has the merit of learning traversability in unsupervised manner, and experiment demo shows the qualitative effectiveness of the method. The scientific contribution is incremental, on the other hand the evaluation and application on real hardware looks promising but lacking quantitive analysis. Given the strengths and weaknesses, my review rate is between neutral and weak accept, hoping the authors can address the review concerns.

---

### Official Review · Reviewer_XNej · 2024-07-29
**Review for paper titled "Velociraptor: Leveraging Visual Foundation Models for Label-Free, Risk-Aware Off-Road Navigation"**

**Originality:** 3
**Technical Quality:** 3
**Clarity Of Presentation:** 4
**Potential Impact:** 3
**Recommendation:** 3
**Confidence:** 5

**Review:**

This paper introduces Velociraptor, a framework that leverages pretrained visual foundation models such as SAM and Dino-v2 to extract relevant features for off-road navigation and uses these representations to predict targets such as costmaps, speedmaps, and uncertainty estimates, which are used to evaluate trajectories generated by a classical planner such as MPPI, for high-speed off-road navigation. The proposed approach is compared with other methods such as GANav, ALTER, and geometric-only method and is shown to be better than existing approaches.

Strengths:

The key contribution of this work is a technique to leverage representations from a pretrained VFM, in combination with geometric features, to predict max speeds, uncertainties, and costmaps, suitable for high-speed off-road navigation. The idea is interesting and the execution looks promising. Overall, I'm in favor of this work and the authors have done a good job at conveying the message in the paper.

Areas of Improvement:

The evaluations are a bit weak in my opinion, which could have been stronger. To compare Velociraptor with baseline methods, there are 2 tasks the authors have evaluated on - the offline tests and the high-speed test. In off-road environments, there could be more than 1 possible route to the goal, which the MHD metric does not capture. It tells me that the speeds and positions are closer to the expert demonstrated trajectory, but it doesn't say that the trajectories chosen by other methods are necessarily worse. More examples are needed to show that Velociraptor does indeed improve over the baseline.

**Quality Of The Limitations Section:**

3

**Questions For Rebuttal:**

1) Equation 1 is not complete. It seems the assumption is that the expert trajectories and the candidate trajectories rolled out by MPPI are of the same length, since there is no normalization term for different-length trajectories. Correct me if i'm wrong, but I think there should be a term where the costs are normalized by the length of the trajectories.
2) In the IRL formulation, there seems to be no mention of a regularization term for the costs. How did the authors regularize the costs generated by the cost function regressor ? Was it weight_decay, L1 regularization, etc?
3) In section 4, parameter count of the shared backbone FCN needs to be mentioned.
4) Figure 2 is a bit confusing and needs a complete overhaul. Specifically, the FPV visual features are connected to the visual map via an arrow, but it needs to be more descriptive for the reader. Please mention what that arrow signifies (is it perfoming a Lift operation?).
5) In Fig 2, it seems there are multiple timesteps of visual and geometric map frames, but the features themselves are generated by aggregating over time as mentioned in text. a) What is the network architecture used to handle these time series feature maps b) Why feed same information when it has already been aggregated over time ?
6) Need additional info about Datasets 1 and 2- How long was the dataset ? What was the speed profile like? Was the same expert driver used to collect both datasets ?
7) "Long" and "short" corrections on Table 3 needs to be defined in text.

**Robotics Focus:**

4

**Summary Of Paper:**

The main idea of this paper is utilizing pretrained visual foundation models as representations to predict targets such as maximum allowed speed, costmap and uncertainty, suitable for the task of high-speed off-road navigation

**Summary Of Recommendation:**

The paper comes with an interesting idea and shows real-robot hardware execution, which are positives worthy of presentation at CoRL. The only lacking part is that the evaluations are weak, and I'm looking forward to hear what the authors have to say in their rebuttal regarding that.

---

### Author Rebuttal · Authors · 2024-08-10

High-level summary of changes:

1. Added additional analysis of hardware runs (Reviewers XNej, Wz3t)
2. Ran additional experiments to quantify the effect of overfitting and feature entropy (Reviewer XNej, K4L2)
3. Revised equations and figure 2 for additional clarity (Reviewers XNej, Wz3t, K4L2)
4. Added additional description and analysis of uncertainty section (Reviewer K4L2, Wz3t)
5. Made plans to open-source the code for this paper (Reviewer Wz3t)
6. We have also added an additional baseline comparison

We have addressed each reviewer’s comments in more detail (attached to the relevant review)

---

### Decision · Program_Chairs · 2024-09-04

**Decision:**

Accept

**Comment:**

Strengths:

- Reviewers highlight the paper's strength in using VFMs to effectively combine geometric and semantic features for improved map quality.
- Reviewer `Ywjl` praises the well-designed experiments with strong baselines and insightful analysis, particularly showcasing VFM benefits in Figure 4 and 5.
- All reviewers agree the paper is well-written and organized, with Reviewer `Ywjl` emphasizing its easy-to-understand approach and clear presentation of the method.

Weaknesses:

- Reviewers `XNej` and `Wz3t` both express concern about the limited evaluation, highlighting the need for more robust comparisons and analysis of the proposed method's effectiveness.

Overall, this seem a good paper with solid real world experiments. There is only one review leaning towards rejection comments on the novelty of this work, which the authors give a good response in rebuttal phase. The strength of this paper outweighs the limitations and I would recommend accepting.